# The treatment gap and the HIV care continuum for cisgender men who have sex with men in three South African cities: Findings from a biobehavioural survey, 2019

Danielle Giovenco[1]⊕*, Tonderai Mabuto[2]⊕*, Albert Manyuchi[3], Helen Savva[4], Anne McIntyre[5], Adrian Puren[6], Griffiths Kubeka[2], Jacqueline Pienaar[2], Cheryl Dietrich[7], Helen Struthers[3,8], Don Operario[1]‡, Eduard J. Sanders[2,9]‡

1 Rollins School of Public Health, Emory University, Atlanta, Georgia, United States of America, 2 The Aurum Institute, Johannesburg, South Africa, 3 Anova Health Institute, Cape Town, South Africa, 4 Division of Global HIV & Tuberculosis, U.S. Centers for Disease Control and Prevention, Pretoria, South Africa, 5 Division of Global HIV & Tuberculosis, U.S. Centers for Disease Control and Prevention, Atlanta, Georgia, United States of America, 6 National Institute for Communicable Diseases, Johannesburg, South Africa, 7 School of Public Health, University of Washington, Seattle, Washington, United States of America, 8 Department of Medicine, University of Cape Town, Cape Town, South Africa, 9 University of Oxford, Oxford, United Kingdom

⊕ These authors contributed equally to this work.
‡ DO and EJS also authors contributed equally to this work.
* danielle.giovenco@emory.edu (DG); tmabuto@auruminstitute.org (TM)

## Abstract

### Background

The HIV continuum of care for men who have sex with men (MSM) in South Africa remains inadequately characterised, hindering the tracking of progress towards ending the HIV epidemic. We estimated HIV prevalence and care continuum indicators for MSM in three South African cities.

### Methods and findings

MSM were recruited via respondent-driven sampling (RDS) methods in Cape Town, Johannesburg, and Mahikeng. Eligibility criteria included age ≥ 18 years, assigned male sex at birth, recent oral or anal sex with a man, and living, working, or socialising in one of the selected cities within the past six months. Participants completed a survey, and dried blood spots were collected to test for HIV antibodies, antiretrovirals, and HIV viral load. RDS weights were used to estimate HIV prevalence and 95-95-95 care continuum indicators. From May to October 2019, 1,790 cisgender MSM were sampled. HIV prevalence was highest in Johannesburg (weighted prevalence = 40.7%, 95% confidence interval 34.4–47.3), followed by Cape Town (25.2%, 20.9–30.1) and Mahikeng (14.7%, 12.0–17.8). Among MSM living with HIV, awareness of status was 67.0% (56.8–75.8) in Cape Town, 67.8% (56.7–77.1) in

**Data availability statement:** All relevant data are within the paper and its Supporting Information files.

**Funding:** This research publication has been supported by the President's Emergency Plan for AIDS Relief (PEPFAR) through the U.S. Centers for Disease Control and Prevention (GGH001981-03 to TM and AM). The findings and conclusions in this publication are those of the authors and do not necessarily represent the official position of the funding agencies.

**Competing interests:** The authors have declared that no competing interests exist.

Johannesburg, and 60.2% (49.9–69.8) in Mahikeng. ART coverage among those aware of their status was 65.1% (53.4–75.2) in Cape Town, 77.9% (67.2–85.8) in Johannesburg, and 72.5% (58.6–83.0) in Mahikeng. Viral suppression among those on ART was 79.3% (59.5–90.9) in Cape Town, 88.7% (77.8–94.6) in Johannesburg, and 90.7% (78.1–96.4) in Mahikeng.

## Conclusions

Achievements towards HIV care indicators were sub-optimal for MSM in three South African sites, revealing potential gaps in the reach and uptake of HIV testing and treatment services. Research identifying multi-level determinants of these gaps is needed to guide the development of contextually appropriate and effective interventions.

## Introduction

South Africa has the largest proportion of people living with HIV globally, with an estimated 7.6 million of the 39.5 million people estimated to be living with HIV in 2022 [1], and an estimated HIV prevalence of 12.7% across all age groups and 17% among adults aged 15–49 years [2]. A key goal of HIV programs globally is to ensure that by the close of 2025, 95% of people living with HIV are aware of their status (1st 95), 95% of those who are aware of their status are receiving antiretroviral treatment (ART) (2nd 95), and 95% of those who are on ART achieving viral suppression (3rd 95) [3]. South Africa has made considerable advancements in pursuit of these targets. In 2022, among all people living with HIV in South Africa, an estimated 89.6% were aware of their status, 90.7% of those who were aware of their status were on ART, and 93.9% of those who were on ART were virally suppressed [2].

South African men have been found to have poorer HIV treatment outcomes compared to women, including being less likely to know their status and achieve viral suppression [4]. In a meta-analysis reporting the proportion of men along the HIV continuum of care in sub-Saharan Africa (SSA), approximately half of men reported not knowing their status (0.49, 95% confidence interval [CI], 0.41–0.58) or not being on treatment (0.58, 95% CI, 0.51–0.65), while over three-quarters of men achieved viral suppression on treatment (0.79, 95% CI, 0.77–0.81) [5]. Engagement in care may be even more challenging among men who have sex with men (MSM), a population with more than twice the HIV prevalence of men in the general population [6]. A 2020 systematic review and meta-analysis of HIV testing and care engagement among MSM in Africa showed that among all MSM living with HIV, 51% (95% credible interval [CrI]: 30–72) knew their status, 73% (95% CrI: 47–88%) were on ART, and 69% (95% CrI: 38–89) were virally suppressed [7].

In South Africa, recent HIV prevalence estimates among MSM are often combined with other groups, such as transgender women (TGW) [6,8]. For example, a 2017 respondent-driven sampling survey of MSM and TGW women in Johannesburg found that 38% of MSM and TGW in Johannesburg were living with HIV. In addition, 57%

(95% CI: 40–73%) knew their status, 30% (95% CI: 17–43%) were currently on ART, and 47% (95% CI: 32–62%) were virally suppressed [9]. Recent estimates of HIV prevalence and care indicators for cisgender MSM in South Africa are limited. The South Africa Men's Health Monitoring Study (SAMHMS 2019) conducted its second round of a bio-behavioural survey among MSM in South Africa, sampling from three of eight cities/towns that were included in the first round, including Cape Town Metropolitan City (Western Cape Province), Johannesburg Metropolitan City (Gauteng Province), and Mahikeng City (capital of the North West Province). The aim of this survey was to estimate HIV prevalence, HIV care indicators, and 95-95-95 progress gaps for cisgender MSM in these three South African cities.

## Methods

### Survey setting and design

A cross-sectional bio-behavioural survey was administered from May to October 2019 among MSM in three South African cities—Cape Town, Johannesburg, and Mahikeng. Cape Town and Johannesburg have large urban support programmes for MSM, whereas Mahikeng is a smaller and peri-urban city with a population of approximately 300,000 and has few support groups and specialized resources for MSM.

Respondent-driven sampling (RDS) was used to recruit MSM [10]. Before commencing the survey, a formative assessment was conducted in all three cities to identify initial survey participants to start the RDS chains of recruitment (i.e., seeds). The seeds were MSM who were well-known in the different communities and were within large social networks of other MSM. In addition, seeds were selected to reflect diversity in age, education level, area of residence in the survey city, ethnicity, known HIV status, and illicit substance use practices. MSM who were willing to serve as seeds were screened for eligibility, provided written informed consent, and completed all survey procedures before each received five coupons to commence the recruitment chains (S2 Appendix, page 3). The coupons had contact information for the survey site, hours and days of survey site operation, and compensation amounts for survey participation and successful referrals of other MSM.

### Sample size and participants

It was determined that 840 participants in Cape Town, 543 participants in Johannesburg, and 895 participants in Mahikeng would be sufficient to estimate the proportion of MSM living with HIV who were virally suppressed with 5% precision around the estimate in each city. A design effect of 1.5 was assumed for our survey, based on previous RDS surveys among MSM in Uganda and South Africa, which reported design effects ranging from 1.20 to 4.65 with a mean of 1.87. This assumption was considered reasonable due to the similarity in design and measures, allowing for a feasible sample size across multiple locations in South Africa. Further, viral load suppression was assumed to be 90% in all three cities; a non-response rate of 3% (i.e., unsuccessful collection of biological specimens, shipping and laboratory issues, and indeterminate results); and HIV prevalence among MSM of 24.8% in Cape Town, 37.0% in Johannesburg, and 18.3% in Mahikeng based on an estimated increase of 10% from a previous survey round conducted in 2016.

Individuals in the three cities were eligible to participate in the survey if they had a valid survey recruitment coupon (described later); were aged 18 years or older; assigned male sex at birth; had engaged in oral or anal sex with a man in the previous six months; and lived, worked, or socialized in the selected cities in the previous six months (S1 Appendix, page 1). Although these criteria included TGW, our analysis was restricted to cisgender MSM as care indicators because estimates for transgender women have been previously reported [11]. Further, eligible individuals who were willing to participate in the survey were asked to consent to complete a behavioural questionnaire (S1 Appendix, page 3), provide blood specimens for laboratory-based tests, and receive results of tests performed. Survey information sheets and consent forms were available in isiZulu, Xhosa, Afrikaans, Sesotho, Setswana, and English.

This work was reviewed by the CDC, deemed not research, and conducted consistent with applicable federal law and CDC policy. In addition, the survey was reviewed and approved by the Human Research Ethics Committee of the University of the Witwatersrand in Johannesburg, South Africa. All participants provided written informed consent for both questionnaire administration and blood specimen collection.

## Survey procedures

Following enrolment, participants completed an interviewer-administered questionnaire adapted for MSM in South Africa and aligned with national programme priorities and performance indicators. The questionnaire was administered in a private room and included questions on sociodemographic characteristics; behaviours related to acquisition of HIV and other sexually transmitted infections (including use of condoms and lubricants, alcohol use, and substance use); access to and utilisation of health services; and experiences of stigma and discrimination related to being MSM (S1 Appendix, page 3). Next, participants were offered and separately consented to receive on-site point-of-care (POC) HIV rapid testing. POC-HIV testing followed the South African national testing guidelines [12], which included counselling before and after the test and using a serial algorithm for HIV diagnosis. Participants were first screened for HIV using the Abon HIV 1/2/O Triline Rapid test (Abon Biopharm, Hangzhou, China). All reactive screening results were confirmed using the First Response HIV1-2.0 Card test (Premier Medical Corporation Private Limited, Mumbai, India). Participants with confirmed HIV-positive results were referred to local organisations or public sector clinics that provided MSM-friendly services for ART initiation and additional clinical support.

This survey protocol included the collection of blood specimens for detecting antibodies to HIV, antiretroviral (ARV) drugs, and viral load measurements. Nurses collected whole blood in tubes with ethylenediaminetetraacetic acid (EDTA) anticoagulant and spotted 75 µl of whole blood in each of the five dotted circles of Whatman 903 filter paper to prepare dried blood spot (DBS) samples. The DBS samples were dried overnight at room temperature and separated by glassine papers to avoid cross-contamination. After drying, they were stored in zip-lock bags with desiccant and humidity indicators and shipped from survey sites at least once a week to the National Institute for Communicable Diseases (NICD) laboratory in Johannesburg, where the tests were conducted. Shortly after providing blood specimens, each participant received 170 South African rand (ZAR) (about 12 United States Dollar (USD)) as compensation for survey participation and travel to the survey site. After that, survey participants were given five coupons to continue the chain of recruitment. For each eligible individual referred to the survey site and successfully enrolled, the referring participant received a secondary compensation of 30 ZAR (about 2 USD) as a supermarket voucher.

## Laboratory testing

Tests for HIV antibodies were performed at NICD using an algorithm of two enzyme immunoassays (the Genscreen Ultra HIV Ag-Ab – Test 1; and Diasorin Murex HIV Ag/Ab Combo – Test 2) and a Western Blot assay (GS HIV -1 Western Blot – Test 3). Samples that were non-reactive on Test 1 were reported as HIV-negative, and samples that were reactive on Test 1 were re-tested using Test 2. The Western Blot test was used to confirm samples that were reactive on both Test 1 and Test 2 and for those that were reactive on Test 1 and non-reactive on Test 2 (discordant results). Specimens with intermediate results on Western Blot were referred for qualitative total nucleic acid (TNA) analysis. If HIV RNA or DNA was detected on the TNA assay, the results were interpreted as positive.

HIV viral load testing was performed for laboratory-confirmed HIV-positive specimens to determine the level of viraemia. Testing was done on DBS samples using the Abbott m2000 HIV Real-Time System (Abbott Molecular Inc., Des Plaines, IL, USA). The presence of ARV drugs (nevirapine, efavirenz, or lopinavir) in the DBS samples was determined using validated in-house, high-performance liquid chromatography (HPLC) coupled with tandem mass spectrometry. This test used an Applied Biosystems API 4000 tandem mass spectrometer (Foster City, CA, USA) in the multiple reaction monitoring (MRM) detection mode for each drug using appropriate MRM transitions. No observable interference in

detecting one drug by the others was anticipated due to the high specificity of this technology and the high degree of validation testing. The limit of detection was set as low as possible (i.e., 0.2 μg/mL) for each of the drugs while maintaining an appropriate signal-to-noise ratio of at least 5:1 for all the drugs. Samples were classified as positive for ARVs if nevirapine, efavirenz, or lopinavir were detected.

## Return of results

The return of laboratory HIV test results was not mandatory, except in cases where the laboratory results differed from the POC-HIV results. In these instances, participants would be contacted by the study nurse to visit the survey site to receive their laboratory results, counselling, and referrals for additional clinical support. Additionally, the return of viral load or ARV drug results was not mandatory. However, those who wanted to receive their results were asked to indicate this at the time of specimen collection and were contacted by the study nurse when the results became available. They could then visit the survey site to receive their laboratory results and counselling if necessary.

## Measures

The primary outcomes were HIV prevalence and the 95-95-95 HIV care continuum indicators.[16] The indicators described the proportion of all MSM who had a laboratory-confirmed HIV-positive test and were aware of their HIV status (first 95); the proportion of MSM who knew their HIV-positive status (i.e., the first 95 numerator) and were on ART, as indicated by the presence of an ARV drug in the DBS sample (second 95); and the proportion of MSM who were on ART (i.e., the second 95 numerator) and had a viral load <1000 copies/mL, as indicated by DBS viral load testing (third 95). At the time of the survey, South African ART guidelines defined viral failure as two viral load results ≥1000 copies/mL [3]. For this study, we used <1000 copies/mL as the threshold to define viral suppression to align with programmatic definitions in South Africa [13] and to ensure comparability with other bio-behavioural surveys [11]. We adjusted self-reported HIV status by incorporating ARV analysis and viral load testing results. Specifically, participants with laboratory-confirmed HIV antibodies and detectable ARV drugs, who self-reported being unaware of their HIV status or HIV-negative, were reclassified as aware of their status.

RDS Analyst (RDS-A), an R-based software package for analysing RDS data, was used to create survey weights inversely proportional to participants' self-reported social network size. For example, participants with a small social network size were less likely to receive a coupon and were assigned a higher weight. In contrast, individuals with a larger social network size had a greater chance of receiving a coupon and were assigned lower weights. We included a series of detailed questions about participants' social network size to improve clarity and accuracy. This approach was informed by the survey team's experience from a previous bio-behavioural survey among female sex workers, where participants struggled to grasp the concept of social networks. We started by asking participants "How many MSM do you know by sight and/or name and who also know you by sight and/name in this city?" (Question 1); and "Of these MSM, how many are 18 years of age and older and live or work or socialise in this city?" (Question 2). Following the second question, the survey staff provided an explanation of the term "personal networks" as those individuals with whom the participants interacted the most in terms of time spent talking and socializing. In the final question (Question 3), participants were then asked to specify the number of people from their response to Question 2 that they would consider to be a part of their personal network (S1 Appendix, page 7).

## Data management and statistical analysis

Survey sites used an electronic fingerprint scanner that generated unique codes to prevent duplicate enrolments. Data from the coupon log books, survey questionnaires, POC-HIV testing results, and laboratory tests were captured using IBM Clinical Development Software, an integrated web-based data management system (https://www.ibm.com/za-en/marketplace/clinical-development). These data were merged and cleaned in STATA (Version 15, College Station, TX),

then exported to RDS-A to run recruitment diagnostic assessments (including recruitment trees, convergence, and bottleneck plots) required to assess convergence and ability to meet RDS estimator assumptions. Gile's successive sampling weights were also generated in RDS-A using population size estimates (PSE) from the first survey rounds in these cities (Cape Town-29,900 MSM; Johannesburg-37,500 MSM; Mahikeng-3,800 MSM) [11]. The sample weights were exported to STATA and used in adjusting sample data to population-level estimates.

We then conducted a descriptive analysis of participant demographic characteristics, HIV prevalence, and the 95-95-95 care indicators separately for Cape Town, Johannesburg, and Mahikeng. HIV prevalence and the care indicators were also described within strata of participant demographic characteristics for each site. We then calculated the "gap" at each step of the care indicators, defined as the proportion of MSM missing from each step if all 95-95-95 targets were reached. Thus, the "treatment gap" was defined as the gap in the proportion of MSM who are living with HIV and are aware of their status who needed to be on treatment to reach the second 95 target. All findings are presented unweighted and weighted, and weighted results are discussed. All analyses were conducted in R Software (Version 2023.06.2 + 561). The findings of this survey are reported according to the guidelines for Strengthening the Reporting of Observational Studies in Epidemiology for RDS studies (STROBE-RDS) to ensure comprehensive and accurate reporting [14].

## Results

Recruitment started with six seeds in Cape Town, 10 in Johannesburg, and 13 in Mahikeng. In Cape Town, 797 of the 2,595 issued coupons (30.7%) were returned to the site by peer recruits. In Johannesburg, 707 of 2,455 issued coupons (28.8%) were returned to the site by peer recruits. In Mahikeng, 896 of 1,894 issued coupons (47.3%) were returned to the site by peer recruits (S2 Appendix, page 1). The recruitment chains were monitored by the number of successive rounds of participant recruitment, starting with the seeds and extending through subsequent referrals (i.e., waves), and the majority of participants were recruited in wave five or higher (66% Cape Town, 60% Johannesburg, and 68% Mahikeng) (Table 1). Overall, 2,145 of 2,400 peer recruits (89%) screened were eligible for survey participation, and all consented to participate. This resulted in 737 participants enrolled in Cape Town, 604 in Johannesburg, and 804 in Mahikeng. Of the 2,145 enrolled in the survey, 355 (16%) described their gender identity as "transgender" (n = 112), "woman" (n = 57), "other" (n = 185), or "don't know" (n = 1) and were excluded from this analysis (S2 Appendix, pages 1–2). The final analysis included 663 cisgender MSM from Cape Town, 379 cisgender MSM from Johannesburg, and 748 cisgender MSM from Mahikeng.

The median age of MSM was similar in Johannesburg and Cape Town (29–30 years), while Mahikeng had a younger MSM population (25 years) (Table 2). MSM in Johannesburg and Mahikeng had similar racial demographic profiles where almost all participants were black. In contrast, more variability in the racial demographic profile was found among MSM in Cape Town where56.5% were black, 35.2% were coloured (mixed race), and 8.3% were of other races. Further, few MSM in Cape Town had completed secondary education (41.9%), compared to Johannesburg and Mahikeng where the majority (78.9% and 80.2%, respectively) had completed secondary education or higher. Estimates of the proportion of MSM who had any income in the past month were low, ranging from 22.2% in Cape Town to 52.0% in Johannesburg.

HIV prevalence (Fig 1; Table 3) was highest in Johannesburg (40.7%, 95% CI 34.4%–47.3%), followed by Cape Town (25.2%, 95% CI 20.9%–30.1%) and Mahikeng (14.7%, 95% CI 12.0%–17.8%). In Cape Town, 67.0% (95% CI

**Table 1. Waves of survey enrolment for cisgender men who have sex with men living in Cape Town, Johannesburg, and Mahikeng, South Africa, May – October 2019.**

|  | Total | Wave 0 | Wave 1 | Wave 2 | Wave 3 | Wave 4 | Wave 5 | Wave 6 | Wave 7 | Wave 8 | Wave 9 | Wave 10 | >Wave 10 |
|---|---|---|---|---|---|---|---|---|---|---|---|---|---|
| Cape Town | 737 | 6 | 27 | 48 | 83 | 88 | 66 | 83 | 95 | 63 | 58 | 50 | 70 |
| Johannesburg | 604 | 10 | 47 | 59 | 51 | 74 | 76 | 70 | 84 | 69 | 47 | 16 | 1 |
| Mahikeng | 804 | 13 | 53 | 73 | 64 | 55 | 51 | 45 | 71 | 72 | 47 | 42 | 218 |

**Table 2. Sociodemographic characteristics of cisgender men who have sex with men living in Cape Town, Johannesburg, and Mahikeng, South Africa, May – October 2019.**

| Characteristic | Unweighted estimates | | | Weighted estimates | | |
|---|---|---|---|---|---|---|
| | Cape Town[a] N=663 | Johannesburg[a] N=379 | Mahikeng[a] N=748 | Cape Town[b] N=28,036 | Johannesburg[b] N=24,189 | Mahikeng[b] N=3,570 |
| Age (years) | 29 (23–37) | 29 (23–34) | 25 (22–30) | 30 (29–32) | 29 (28–31) | 25 (25–26) |
| Age group | | | | | | |
| 18-24 years | 197/663 (29.7%) | 123/379 (32.5%) | 342/748 (45.7%) | 29.4% (24.2%–35.1%) | 31.9% (26.1%–38.3%) | 44.6% (40.5%–48.9%) |
| 25-34 years | 255/663 (38.5%) | 163/379 (43.0%) | 295/748 (39.4%) | 38.5% (33.0%–44.3%) | 39.7% (33.5%–46.3%) | 37.8% (33.8%–42.0%) |
| 35+years | 211/663 (31.8%) | 93/379 (24.5%) | 111/748 (14.8%) | 32.2% (27.2%–37.6%) | 28.4% (22.6%–35.0%) | 17.6% (14.3%–21.4%) |
| Race | | | | | | |
| Black | 397/663 (59.9%) | 362/379 (95.5%) | 712/748 (95.2%) | 56.5% (50.6%–62.2%) | 94.9% (89.7%–97.5%) | 95.8% (93.9%–97.2%) |
| Coloured | 216/663 (32.6%) | 12/379 (3.2%) | 35/748 (4.7%) | 35.2% (29.8%–41.1%) | 3.4% (1.5%–7.6%) | 4.1% (2.8%–6.0%) |
| Other | 50/663 (7.5%) | 5/379 (1.3%) | 1/748 (0.1%) | 8.3% (5.6%–12.0%) | 1.8% (0.4%–7.0%) | 0.0% (0.0%–0.3%) |
| Country of birth | | | | | | |
| South Africa | 586/663 (88.4%) | 326/379 (86.0%) | 738/748 (98.7%) | 87.9% (83.4%–91.4%) | 81.6% (75.7%–86.3%) | 98.8% (97.4%–99.5%) |
| Other | 77/663 (11.6%) | 53/379 (14.0%) | 10/748 (1.3%) | 12.1% (8.6%–16.6%) | 18.4% (13.7%–24.3%) | 1.2% (0.5%–2.6%) |
| Highest level of education | | | | | | |
| Less than secondary | 307/661 (46.4%) | 70/379 (18.5%) | 139/746 (18.6%) | 58.1% (52.5%–63.6%) | 21.1% (16.2%–27.0%) | 19.8% (16.5%–23.6%) |
| Secondary | 272/661 (41.1%) | 264/379 (67.0%) | 455/746 (61.0%) | 33.4% (28.4%–38.7%) | 69.0% (62.6%–74.7%) | 62.8% (58.6%–66.8%) |
| Tertiary | 82/661 (12.4%) | 45/379 (14.5%) | 152/746 (20.4%) | 8.5% (6.0%–11.9%) | 9.9% (6.8%–14.2%) | 17.4% (14.6%–20.6%) |
| Income in past month | | | | | | |
| No monthly income | 498/663 (75.1%) | 181/379 (47.8%) | 510/748 (68.2%) | 77.8% (72.6%–82.3%) | 48.0% (41.5%–54.6%) | 67.1% (63.0%–71.0%) |
| Any monthly income | 165/663 (24.9%) | 198/379 (52.2%) | 238/748 (31.8%) | 22.2% (17.7%–27.4%) | 52.0% (45.4%–58.5%) | 32.9% (29.0%–37.0%) |

[a] median (interquartile range) or n/N (%)

[b] median (95% confidence internal) or % (95% confidence interval)

56.8%–75.8%) of MSM living with HIV were estimated to know their positive status (first 95; Fig 2); this was 67.8% (95% CI 56.7%–77.1%) in Johannesburg, and 60.2% (95% CI 49.9%–69.8%) in Mahikeng. Among those who knew their HIV status, 65.1% (95% CI 53.4%–75.2%) in Cape Town, 77.9% (95% CI 67.2%–85.8%) in Johannesburg, and 72.5% (95% CI 58.6%–83.0%) in Mahikeng were on ART (second 95; Fig 2). Of those on ART, 79.3% (95% CI 59.5%–90.9%) in Cape Town, 88.7% (95% CI 77.8%–94.6%) in Johannesburg, and 90.7% (95% CI 78.1%–96.4%) in Mahikeng were virally suppressed (third 95; Fig 2). A description of HIV prevalence and 95-95-95 care indicators with additional stratification by demographic characteristics is provided in S2 Appendix on pages 8–13.

Estimates for reaching the target for the first 95 (Fig 3) were similar in Cape Town (28.0% gap in the proportion of MSM who are living with HIV who need to become aware of their status to reach 95%) and Johannesburg (27.2% gap) compared with Mahikeng (34.8% gap). The gap to reaching the second 95 was lowest in Johannesburg (37.5%), and similar for Cape Town (46.7%) and Mahikeng (46.6%). The gap for reaching the third 95 was highest in Cape Town (51.2%), followed by Mahikeng (46.2%), and Johannesburg (38.9%).

## Discussion

This descriptive survey reveals substantial gaps in awareness of HIV status, current ART use, and viral suppression among MSM in three South African cities. Mahikeng, the smallest of the three cities, had the largest gap (35%) to reaching the 95% target for HIV status awareness (1st 95), whereas both Mahikeng and Cape Town had gaps ranging from 47–51% for ART use (2nd 95) and viral suppression (3rd 95) compared to 38–39% for Johannesburg. These

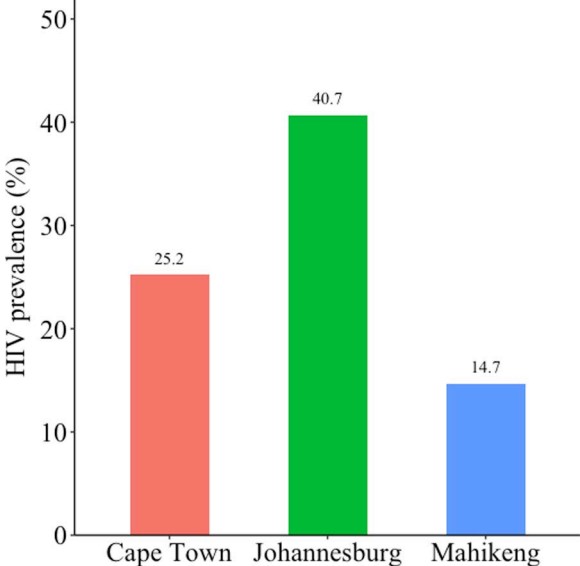

**Fig 1. HIV prevalence estimates for cisgender men who have sex with men living with HIV in Cape Town, Johannesburg, and Mahikeng, South Africa, May – October 2019.**

**Table 3. HIV prevalence and care continuum indicators for cisgender men who have sex with men living in Cape Town, Johannesburg, and Mahikeng, South Africa, May – October 2019.**

|  | Unweighted estimates | | | Weighted estimates | | |
|---|---|---|---|---|---|---|
|  | Cape Town[a] N = 663 | Johannesburg[a] N = 379 | Mahikeng[a] N = 748 | Cape Town[b] N = 28,036 | Johannesburg[b] N = 24,189 | Mahikeng[b] N = 3,570 |
| HIV prevalence | 226/663 (34.1%) | 181/379 (47.8%) | 125/748 (16.7%) | 25.2% (20.9%–30.1%) | 40.7% (34.4%–47.3%) | 14.7% (12.0%–17.8%) |
| Aware of HIV positive status (1st 95) | 157/226 (69.5%) | 135/181 (74.6%) | 74/125 (59.2%) | 67.0% (56.8%–75.8%) | 67.8% (56.7%–77.1%) | 60.2% (49.9%–69.8%) |
| Aware of status and on ART (2nd 95) | 104/154 (67.5%) | 105/134 (78.4%) | 53/73 (72.6%) | 65.1% (53.4%–75.2%) | 77.9% (67.2%–85.8%) | 72.5% (58.6%–83.0%) |
| On ART and virally suppressed (3rd 95) | 92/104 (88.5%) | 94/105 (89.5%) | 48/53 (90.6%) | 79.3% (59.5%–90.9%) | 88.7% (77.8%–94.6%) | 90.7% (78.1%–96.4%) |

[a]n/N (%)

[b]% (95% confidence interval)

estimates of care continuum indicators for all three cities are far below the UNAIDS 95-95-95 goals [15]. HIV prevalence data also showed differences between the cities, with Johannesburg having a higher prevalence than both Cape Town and Mahikeng.

This represents the most current description of HIV prevalence and care continuum indicators among MSM in South Africa. Prior RDS research has examined these outcomes among TGW in three South African cities—Buffalo City, Cape Town, and Johannesburg [10]. This survey found high estimates of HIV prevalence (46–63%) and poor progress across care indicators (24–54% knew their status, 65–82% were on ART, and 34–55% were virally suppressed), revealing substantial treatment gaps for TGW in South Africa as well. Importantly, TGW and MSM are distinct key populations in South Africa, and each group warrants strengthened efforts to ensure they are reached, tested, engaged, and retained in HIV prevention and care programming.

Further research is needed to identify the multilevel determinants of HIV testing and care engagement among South African MSM and inform interventions that improve diagnosis and treatment outcomes. A RDS survey in Johannesburg among MSM and TGW found that viral suppression was associated with older age, neighbourhood, and transactional sex

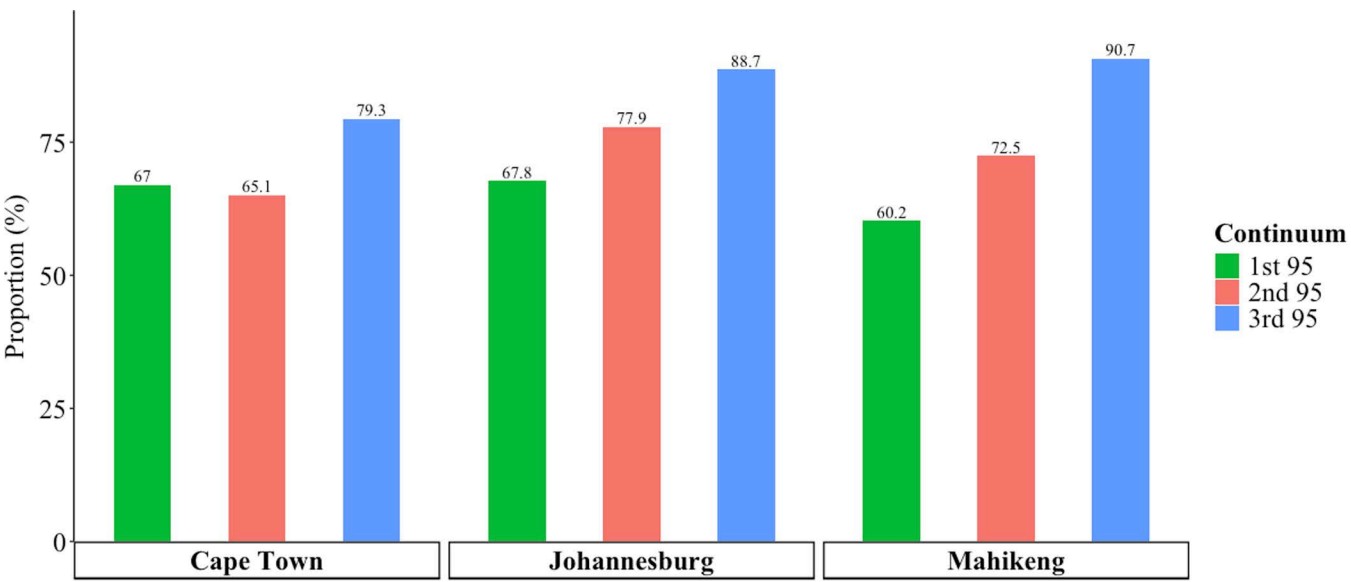

**Fig 2. HIV care continuum indicator estimates for cisgender men who have sex with men living with HIV in Cape Town, Johannesburg, and Mahikeng, South Africa, May – October 2019.** Key: 1st 95 = the estimated proportion of MSM living with HIV who were aware of their HIV-positive status; 2nd 95 = the estimated proportion of MSM who know their status and are on ART; and 3rd 95 = the estimated proportion of MSM who are on ART and virally suppressed.

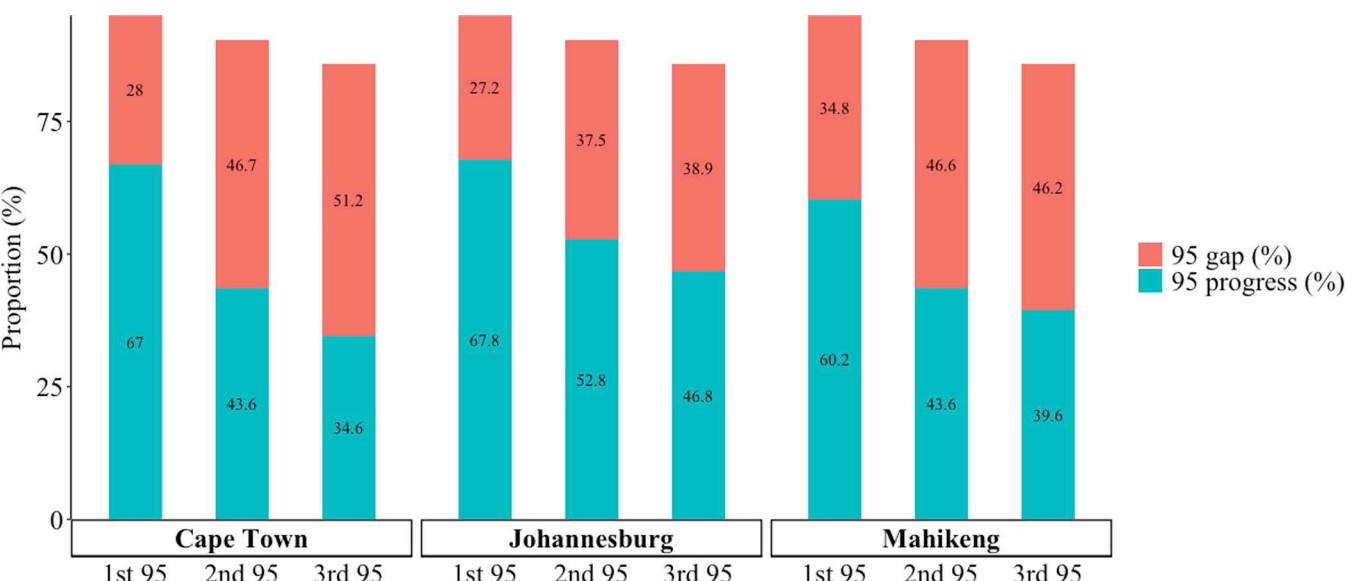

**Fig 3. Progress and gaps towards reaching the 95-95-95 HIV care continuum targets for cisgender men who have sex with men living with HIV in Cape Town, Johannesburg, and Mahikeng, South Africa, May – October 2019.** Key: 1st 95 = the estimated proportion of MSM living with HIV who were aware of their HIV-positive status; 2nd 95 = the estimated proportion of MSM who know their status and are on ART; 3rd 95 = the estimated proportion of MSM who are on ART and virally suppressed; 95 gap = the proportion of MSM missing from each step if all 95-95-95 targets were reached.

[9]. A social-ecological framework can be useful to contextualize the multilevel factors contributing to poor HIV outcomes among MSM in South Africa, including individual behaviours, social relationships, community norms, and structural barriers [16]. For example, MSM living with HIV in South Africa often face stigma related to their HIV status and sexual identity, which can contribute to psychosocial distress and disengagement from care [15,17]. Structural interventions—such as anti-discrimination protections and the promotion of equitable, differentiated HIV testing and treatment models—are critical to closing gaps in the HIV care continuum [18]. Further, strategies like peer mobilization, HIV self-testing, and partner notification have effectively identified undiagnosed infections among MSM and TGW in Kenya [19], while integrated HIV and gender-affirming care models have improved service access and viral suppression among transgender individuals in South Africa [20].

This research has several important limitations. Our methods for qualitative detection of ARVs used to define the 2nd 95 did not include the detection of dolutegravir (DTG)-based regimens, which now form the backbone of first-line ART regimens in South Africa. However, because South African guidelines recommending DTG rollout were only published in October 2019—the same month this study concluded—any underestimation of the 2nd 95 is likely minimal [21]. Additionally, findings from this survey may not be generalizable to MSM in other cities in South Africa or regions in SSA more broadly. Finally, although our descriptive cross-sectional findings provide evidence for gaps in the 95-95-95 targets for MSM living with HIV in three South African cities, this analysis cannot provide data-driven interpretations about the factors determining these treatment gaps. Nevertheless, this is the first survey describing HIV care continuum indicators among South African MSM specifically and contributes to our knowledge of progress towards UNAIDS' targets.

In conclusion, we found high HIV prevalence and sub-optimal care indicators for MSM in three South African cities. Despite progressive legal protections—including same-sex marriage and anti-discrimination policies—many MSM in South Africa remain disengaged from HIV services [15]. To address these gaps, HIV testing and treatment programs may benefit from sustained outreach and targeted service delivery. Given persistent barriers to care, policies and interventions that promote inclusive, non-stigmatizing, and accessible services for MSM are urgently needed [19,22]. Strengthening monitoring and surveillance of the HIV care continuum among MSM may also help accelerate progress towards the UNAIDS 95-95-95 goals.

## Supporting information

**S1 Appendix. Eligibility screening form and survey instrument**
(PDF)

**S2 Appendix. Survey recruitment flow diagrams, RDS recruitment trees, recruitment diagnostics, and supplementary tables**
(PDF)

**S3 Appendix. Minimal dataset**
(CSV)

**S1 Checklist. Human participants research checklist**
(PDF)

## Author contributions

**Conceptualization:** Tonderai Mabuto, Albert Manyuchi, Helen Savva, Anne McIntyre, Adrian Puren, Jacqueline Pienaar, Helen Struthers.

**Data curation:** Tonderai Mabuto, Albert Manyuchi, Helen Savva, Anne McIntyre, Adrian Puren, Griffiths Kubeka, Cheryl Dietrich, Helen Struthers.

**Formal analysis:** Danielle Giovenco, Tonderai Mabuto.

**Investigation:** Tonderai Mabuto, Anne McIntyre, Adrian Puren.

**Methodology:** Danielle Giovenco, Tonderai Mabuto, Anne McIntyre, Adrian Puren, Don Operario, Eduard J. Sanders.

**Project administration:** Tonderai Mabuto.

**Validation:** Tonderai Mabuto, Albert Manyuchi, Anne McIntyre.

**Writing – original draft:** Danielle Giovenco, Tonderai Mabuto, Don Operario, Eduard J. Sanders.

**Writing – review & editing:** Danielle Giovenco, Tonderai Mabuto, Albert Manyuchi, Helen Savva, Anne McIntyre, Adrian Puren, Griffiths Kubeka, Jacqueline Pienaar, Cheryl Dietrich, Helen Struthers, Don Operario, Eduard J. Sanders.

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
