## [Decision Letter · Decision Letter 0]

21 Apr 2025

PGPH-D-24-02814

The treatment gap and the HIV care continuum for cisgender men who have sex with men in three South African cities: Findings from a biobehavioural survey, 2019

Dear Dr. Giovenco,

Thank you for submitting your manuscript to PLOS Global Public Health. After careful consideration, we feel that it has merit but does not fully meet PLOS Global Public Health’s publication criteria as it currently stands. Therefore, we invite you to submit a revised version of the manuscript that addresses the points raised during the review process.

We look forward to receiving your revised manuscript.

Kind regards,

Max Carlos Ramírez-Soto, BSc, MPH, PhD, FRSPH, FECMM

Academic Editor

Journal Requirements:

2. Please provide separate figure files in .tif or .eps format.

3. We noticed that you used “unpublished data" in the manuscript. We do not allow these references, as the PLOS data access policy requires that all data be either published with the manuscript or made available in a publicly accessible database. Please amend the supplementary material to include the referenced data or remove the references.

Additional Editor Comments (if provided):

Reviewers' comments:

Reviewer's Responses to Questions

**Comments to the Author**

1. Does this manuscript meet PLOS Global Public Health’s publication criteria ? Is the manuscript technically sound, and do the data support the conclusions? The manuscript must describe methodologically and ethically rigorous research with conclusions that are appropriately drawn based on the data presented.

Reviewer #1: Yes

Reviewer #2: Yes

2. Has the statistical analysis been performed appropriately and rigorously?

Reviewer #1: Yes

Reviewer #2: Yes

3. Have the authors made all data underlying the findings in their manuscript fully available (please refer to the Data Availability Statement at the start of the manuscript PDF file)?

Reviewer #1: Yes

Reviewer #2: Yes

4. Is the manuscript presented in an intelligible fashion and written in standard English?

Reviewer #1: Yes

Reviewer #2: Yes

5. Review Comments to the Author

Reviewer #1: The manuscript is well written, and it reads well. There are no discrepancies in the statistical analysis. The results are well presented, and I do believe this manuscript meets PLOS one standards. Thank you.

Reviewer #2: Thank you for the opportunity to review this important and timely manuscript. Overall, the manuscript is well written. The methods are well described and easy to follow. The comments below are offered in the spirit of collegiality to strengthen the manuscript.

Introduction -

- To frame SA’s HIV epidemic please provide the case rate rather than incidence so that readers can contextualize within the global HIV epidemic.

- It seems a bit misleading in the second sentence to frame 95%-95%-95% as a country specific HIV program goal rather than a global UNAID initiative particularly when neighboring countries have nearly achieved these goals.

- In line 29, please specify where these studies are done, in SA? Globally? Also which HIV outcomes are the authors referencing?

- Line 30 - the authors have not defined SSA

Methods

- Why was undetectable defined as vl<1000? At the time the study was conducted, I believe the South African ART guidelines defined undetectable as <400. This requires an explanation.

Results

- Given that 707 issued coupons in Joburg were returned and none declined to participate, how were so many excluded from Joburg to include a sample of only 379? Were most /all TGW and gender non-conforming individuals there? If so, please state.

Discussion

- The third paragraph of the discussion needs more clarity. Reviewing the related study about factors associated with VS among MSM is prudent, but the authors go on to discuss the social ecological model (which is not an intersectional model) then go on to very casually toss in discussion about intersectional stigma and its multifaceted manifestations, syndemic theory, and structural discrimination. These are all huge topics that should not be unwrapped for a single toss away sentence. The authors should either contend with these theories and constructs appropriately or stick to the social ecological model alone.

- Line 308 the authors bring up an important consideration about dolutegravir. To my recollection, SA began using dolutegravir as a second line agent for suspected treatment failure in October/November 2019 and may be more influential than authors suggest. Did the survey ask any questions about ART regimens or regimen switches that may help to assure that this is not a more significant limitation?

6. PLOS authors have the option to publish the peer review history of their article (what does this mean? ). If published, this will include your full peer review and any attached files.

**Do you want your identity to be public for this peer review?** For information about this choice, including consent withdrawal, please see our Privacy Policy .

Reviewer #1: No

Reviewer #2: No

---

## [Editor Report · Decision Letter 1]

17 Jun 2025

The treatment gap and the HIV care continuum for cisgender men who have sex with men in three South African cities: Findings from a biobehavioural survey, 2019

PGPH-D-24-02814R1

Dear Dr. Giovenco,

We are pleased to inform you that your manuscript 'The treatment gap and the HIV care continuum for cisgender men who have sex with men in three South African cities: Findings from a biobehavioural survey, 2019' has been provisionally accepted for publication in PLOS Global Public Health.

Best regards,

Max Carlos Ramírez-Soto, BSc, MPH, PhD, FRSPH, FECMM

Academic Editor

https://orcid.org/0000-0003-0471-6746

No comments